# Recent Insights into Particulate Matter (PM_2.5_)-Mediated Toxicity in Humans: An Overview

**DOI:** 10.3390/ijerph19127511

**Published:** 2022-06-19

**Authors:** Prakash Thangavel, Duckshin Park, Young-Chul Lee

**Affiliations:** 1Department of BioNano Technology, Gachon University, 1342 Seongnamdaero, Sujeong-gu, Seongnam-si 13120, Gyeonggi-do, Korea; prakashjacob47@gmail.com; 2Korea Railroad Research Institute (KRRI), 176 Cheoldobakmulkwan-ro, Uiwang-si 16105, Gyeonggi-do, Korea; 3Well Scientific Laboratory Ltd., 305, 3F, Mega-Center, SKnTechnopark, 124 Sagimakgol-ro, Jungwon-gu, Seongnam-si 13207, Gyeonggi-do, Korea

**Keywords:** air pollution, particulate matter, PM_2.5_, health effects, COVID, H1N1, SARS

## Abstract

Several epidemiologic and toxicological studies have commonly viewed ambient fine particulate matter (PM_2.5_), defined as particles having an aerodynamic diameter of less than 2.5 µm, as a significant potential danger to human health. PM_2.5_ is mostly absorbed through the respiratory system, where it can infiltrate the lung alveoli and reach the bloodstream. In the respiratory system, reactive oxygen or nitrogen species (ROS, RNS) and oxidative stress stimulate the generation of mediators of pulmonary inflammation and begin or promote numerous illnesses. According to the most recent data, fine particulate matter, or PM_2.5_, is responsible for nearly 4 million deaths globally from cardiopulmonary illnesses such as heart disease, respiratory infections, chronic lung disease, cancers, preterm births, and other illnesses. There has been increased worry in recent years about the negative impacts of this worldwide danger. The causal associations between PM_2.5_ and human health, the toxic effects and potential mechanisms of PM_2.5_, and molecular pathways have been described in this review.

## 1. Introduction

Particulate matter (PM) is made up of solid and liquid particles that are discharged directly into the air as a result of diesel use, road and agricultural dust, and industrial activity. The morphological, chemical, physical, and thermodynamic features of PM are diverse. Because of its size, density, and thermal conditions, as well as wind speed, PM remains suspended in the air, polluting it [1,2]. The WHO reported that around 7 million people die every year due to exposure to polluted air, and that ambient air pollution, especially in low/middle-income countries, caused 4.2 million deaths in 2016 (World Health Organization, 2016). Air pollution is caused by a mixture of substances such as gases, particles, and biological components in the earth’s atmosphere. The toxic effects caused by particle pollution on humans are dependent on their size, superficial area, and chemical composition [3]. The adverse health effects of gaseous pollutants have been extensively studied in the past few decades; however, despite differences in air pollutants and variations in local atmospheres, the basic features of acute and long-term health effects caused by such pollutants are yet to be explored [4]. Epidemiological studies collecting data on concentrations of the main gas pollutants, exposure, and dose for the exposed population have been conducted to determine the specific causes of observed health effects [5]. Air pollution is the single largest environmental causative agent of various diseases. The health effects induced by exposure to air pollution mainly manifest in elderly people with pre-existing cardiopulmonary diseases [6], cerebrovascular diseases [7], neurodegenerative diseases [8], bronchitis [9], emphysema [10], increased irritation of the eye and respiratory system [11], asthma attacks [12], respiratory infections [13], and so on. Air pollution has also been associated with adverse pregnancy outcomes such as preeclampsia and hypertensive disorders [14,15]. PM can be classified by its aerodynamic diameter size as PM_10_ (particles ≤ 10 µm in diameter); PM_2.5_ (particles ≤ 2.5 µm in diameter), also called fine particles; and PM_0.1_ (particles ≤ 0.1 µm in diameter), called ultrafine particles, which have different health effects, as the particles are strongly linked to particle size, which can deposit in the lung and can navigate through bronchioles and escape lung defense mechanisms [16]. These particles are extensively found in the atmosphere and are released by various sources such as dust storms, forest fires, and volcanoes, as well as human activities, including transportation, fuel burning, and industrial processes [17]. The Global Burden of Disease (GBD) study estimated that 5 million deaths are caused by PM_2.5_ annually. PM_2.5_ is characterized by fine particles that have a large surface area. Due to their size, they can accumulate more compared to PM_10_, propagate long distances, become stagnant in the atmosphere, stay in the air longer, and travel farther [18]. PM_2.5_ is composed of primary particles that are emitted directly into the atmosphere and secondary particles produced by chemical reactions between precursor gases [19,20]. Primary PM_2.5_ particles which are directly emitted into the atmosphere can originate from both natural sources, such as dust storms and forest fires, and anthropogenic sources, such as fossil fuel combustion, cigarette smoke, and biomass burning. Secondary PM_2.5_ particles are generated by chemical reactions between PM from anthropogenic and biogenic sources [21,22]. The exact mechanisms associated with the impact of PM_2.5_ on the human body remain unclear. It is hypothesized that inhaled PM accumulates in the lungs and activates inflammatory cells, leading to the release of mediators and the stimulation of alveolar receptors, which causes an imbalance in the autonomic nervous system (ANS) and neuroendocrine pathway [23,24,25]. The other mechanism is translocation of PM via the pulmonary epithelium. The primary pollutants in fine PM enter the blood circulation and affect the whole organism. However, when there is inflammation in the lungs due to PM_2.5_, the inflammation leads to oxidative stress and causes vascular dysfunction [26,27]. Air pollutant particles, such as PM_2.5_, have a negative impact on the productivity of agricultural workers, land miners, and sewer employees due to the particles generated during their work [27]. Mineral dusts, such as those containing free crystalline silica (e.g., as quartz); organic and vegetable dusts, such as wheat, wood, cotton, and tea dust; and pollens may be found in the workplace. Dust particles or additives can enter the airway and impair lung function [28,29]. In this review, we discuss the sources of PM_2.5_ and its health effects on humans based on epidemiological, experimental, and molecular studies.

## 2. Literature Review

A literature search was carried out in the PubMed database (National Library of Medicine) and ScienceDirect for the literature published between January 2010–2022 using the following keywords: particulate matter 2.5, PM_2.5_, PM_2.5_ health effects, PM_2.5_ and respiratory diseases, PM_2.5_ and cardiovascular diseases, PM_2.5_ and cancer, PM_2.5_ and cardiovascular diseases, PM_2.5_ and SARS, H1N1. 

We only included studies that involve PM_2.5_ and health effects, epidemiological studies that involve health impact of exposure of PM_2.5_, studies expressing quantitative exposure-response relationship between PM_2.5_ and health outcomes; health outcomes were hospital utilization due to PM_2.5_ exposure causes such as asthma, COPD, CVD, lung cancer, neurodegenerative diseases and role of PM_2.5_ in SARS and H1N1 virus. The database identified literature was then preselected by following inclusion and exclusion criteria. [A] Inclusion criteria: (1. Duplicate check, 2. Studies involving health outcomes, 3. Studies done on cell lines, group of population involving mortality and morbidity, 4. Hospital outcomes and hospital utilization, 5. Study population involving healthy and non-healthy individuals.) [B] Exclusion criteria: (1. Unrelated studies such as other pollutants, 2. Studies that doesn’t involve health outcomes, 3. Studies that are not published in English, 4. Studies that lacked sufficient information, 5. Studies that were not recently published. An author screened the literature based on the inclusion and exclusion criteria its inclusion in the manuscript. The systematic screening steps are summarized (Figure 1). 

## 3. Sources of PM_2.5_

PM_2.5_ sources and concentrations may vary significantly across locations due to distinct climatic conditions, emission sources, and dispersion patterns. The sources can be either natural or anthropogenic [30]. Depending on the location, different sources may contribute to PM_2.5_ levels, such as vehicle traffic, dust resuspension, biomass burning, power plants, sea salt, industrial emissions, ship emissions, and aircraft emissions. Understanding the sources and effects of high levels of PM_2.5_ is essential to developing effective strategies to control these levels and protect human health [31]. Several source apportionment methods, such as (1) methods based on the evaluation of monitoring data; (2) methods based on emission inventories and dispersion models to simulate aerosol emission, formation, transport, and deposition; and (3) methods based on the statistical evaluation of PM chemical data acquired at receptor sites (receptor models), have been proposed to identify sources of PM_2.5_ and their contributions to air quality [32,33]. PM_2.5_ is mainly composed of various undetermined fractions. It is primarily produced by combustion and emissions of fuel-powered vehicles and the wear and tear of auto parts [34]. The major components in PM_2.5_ are black carbon [35], polycyclic aromatic hydrocarbons [36,37], aryl hydrocarbons [38], volatile organic hydrocarbons [39], heavy metals [40], organic compounds [41], minerals [42], inorganic ions [43], and biological materials [44], which collectively make up at least 79–85% of the total mass [45]. PM can be emitted directly from the main sources or indirectly through the conversion of gaseous emissions in the atmosphere. Studies on the elemental composition of PM_2.5_ have revealed elevated levels of elements, including Al, As, Br, Ca, Cl, Cr, Fr, K, Mg, Mn, Na, Pb, Ti, and Zn, as well as sulfate, nitrate, and ammonium ions [46]. Various sources of PM_2.5_ and their chemical components have been listed (Table 1). Other potential sources of PM_2.5_ include human activities such as residential cooking, smoking, social and economic development, meteorological factors, and secondary generation of air pollution.

Secondary pollutants are generated when primary pollutants react with each other. Examples of secondary pollutants include ozone (O_3_) and secondary organic carbon (SOC), O_3_, which is formed when hydrocarbons (HC) and nitrogen oxides (NOx) react with each other in the presence of sunlight. SOC is generated by photochemical reactions of gaseous precursors from primary organic carbon (POC) [47].

## 4. Health Complications

The GBD study estimated the attributable levels of PM_2.5_ in 195 countries and territories worldwide. Ambient PM_2.5_ and household PM_2.5_ ranked among the top ten leading global risk factors for disease [48]. Exposure to environmental PM_2.5_ has been associated with an increase in the incidence and mortality of many diseases. High risk of PM_2.5_-related death from stroke, ischemic heart disease, chronic obstructive pulmonary disease (COPD), lung cancer, and other diseases around the world has been demonstrated in several studies [49,50,51,52,53,54]. The lungs, the initial sites of PM_2.5_ deposition in the airway, are among the primary targets of PM_2.5_-induced toxicity, which leads to airway inflammation, impairing normal immune responses of the lungs and making them susceptible to various respiratory infections [55]. It’s been hypothesized that PM_2.5_ impairs the normal immune responses by various mechanisms. Firstly, PM_2.5_ can damage the bronchial mucociliary system, reducing bacterial clearance [56]. Secondly, PM_2.5_ and PM_2.5_-induced inflammatory cytokine net disruption may cause the death of lung epithelial cells and fibroblasts, as well as the inhibition of gap junctional intercellular communication between these cells, increasing epithelial barrier permeability and impairing their function as physical barriers for pulmonary innate immunity [57]. Thirdly, alveolar macrophages are essential inflammation regulators and are required for antibacterial activity in the lower airway [58]. Recently, increasing evidence has shown that PM_2.5_ not only inhibits alveolar macrophage phagocytosis by disrupting the normal physical and immunological function of lung surfactants, such as di-palmitoyl-phosphatidylcholine and amino acids related to opsonin proteins [59], but they also impair the response of natural killer (NK) cells [60] and inhibit antibacterial capabilities through transferrin-mediated Fe^3+^ transport [61], disrupting the expression of toll-like receptors (TLRs) and microtubule architecture [62,63]. Recently, increasing evidence has shown that PM_2.5_ not only inhibits alveolar macrophages phagocytosis by disrupting the normal physical and immunological function of lung surfactants, such as di-palmitoyl-phosphatidylcholine and amino acids related to opsonin proteins [59], which generally act as opsonins and enhance alveolar macrophages phagocytosis to bacteria, and impairing the response of natural killer (NK) cells [60], which also enhance alveolar macrophages phagocytosis, but also directly inhibiting alveolar macrophages antibacterial capabilities through a variety of methods. These methods include influencing transferrin-mediated Fe^3+^ transport to alveolar macrophages [61], affecting the expression of toll-like receptors (TLRs); disrupting microtubule architecture; and decreasing their phagocytic activities [62,63]. All of these would lead to reduced pulmonary immunity and facilitate infectious illnesses. Recently, chronic exposure to PM_2.5_ was found to be linked with the development of diabetes mellitus (DM), inducing multiple abnormalities associated with the development of type 2 diabetes mellitus (T2DM), insulin resistance (IR), adipose inflammation, and hepatic endoplasmic reticulum (ER) stress. Alterations in ER stress and inflammatory pathways have been proposed to be the mechanisms by which PM_2.5_ induces IR and T2DM [64]. Furthermore, PM_2.5_ exposure not only leads to subclinical changes in cardiovascular function, but also impairs the function of the cardiac autonomic nervous system (ANS), leading to a decline in heart rate variability, which is inevitably related to cardiovascular morbidity and mortality [65]. Epidemiological evidence suggests that PM_2.5_ is a risk factor for chronic kidney disease (CKD). Moreover, PM_2.5_ leads to a decrease in glomerular filtration rate (GFR) and is related to the prevalence and incidence of CKD [66]. Protecting the environment and the environmental health of mothers and infants remains a top global priority. Epidemiological evidence suggests that maternal PM_2.5_ exposure during pregnancy is associated with negative birth outcomes, including preterm birth, low birth weight, and post neonatal infant mortality [67,68,69,70,71]. The health impacts of PM_2.5_ are summarized (Table 2). In addition, PM_2.5_ influences several other adverse health effects such as bone damage, liver fibrosis, lung cancer, macrosomia, Alzheimer’s disease, ovarian dysfunction, hormone dysregulation, and compromised antiviral immunity [72,73,74,75,76,77,78].

## 5. PM_2.5_ and Airway Inflammation

The airways and lungs, the initial sites of PM_2.5_ entry and deposition, are the primary targets of its toxicity. After PM_2.5_ inhalation, the fine particles deposit on the surface of the airway and pulmonary bronchi and alveoli before being internalized into lung cells such as epithelial cells and alveolar macrophages [98]. Thereafter, PM_2.5_ triggers oxidative stress and impairs the normal function of cells or can even induce apoptosis by different mechanisms, such as autophagy. Furthermore, PM_2.5_ induces inflammatory responses, which play a major role in respiratory damage. Epidemiological evidence shows that PM_2.5_ can regulate different inflammation-related signaling pathways, as indicated by elevated Th2 cytokine (interleukin (IL)-4, IL-5, and IL-13) levels in bronchoalveolar lavage fluid (BALF), increased expression of IL-8, histamines, and leukotriene, and the promotion of eosinophil infiltration, leading to airway hyperresponsiveness [98]. Moreover, PM_2.5_ exposure promotes the release of IL-33, which drives Th2-biased immune responses and upregulates the expression of IL-6 and IL-1β in human bronchial epithelial cells, inducing the occurrence of lung injury by regulating the levels of lipid mediators and sphingosine-1-phosphate [99,100]. Similarly, PM_2.5_ induces oxidative stress-signaling pathways by activating the PI3K/Akt, NF-κb, Nrf2-KEAP1-ARE, JAK/STAT, and MAPK signaling pathways, elevating ROS and Nrf2 levels, increasing nitric oxide synthase and NO generation, and upregulating the expression of IL-6, IL-8, and cyclooxygenase-2 [101,102,103,104]. Moreover, PM_2.5_ exposure may alter and impair the normal immune system by inducing M1 (through enhanced response of CD86, CXCL1, CXCL2, IL-1β, IL-6, NOS2, and TNF-α) and M2 macrophage polarization (by increasing the levels of arginase-1, CD206, and YM-1, and inhibiting histone deacetylase 2) [105,106,107,108]. Exposure to PM_2.5_ also alters immune homeostasis. Epidemiological evidence confirms that after PM_2.5_ exposure, the Th1/Th2 response is broken and the balance shifts toward increased (T cells) Th2 immune responses accompanied by the activation of toll-like receptors (TLR2 and TLR4) and the MYD88/COX-2 signaling pathway [109]. PM_2.5_ exposure also promotes the expression of GATA3 and RUNX3, and reduces the expression of T-bet, inducing a Th2-biased immune response accompanied by increases in the IFN-γ, IL-4, and IL-13 levels, which leads to immune system imbalances [110,111]. Exposure to PM_2.5_ impairs the differentiation of Treg cells and promotes the differentiation of Th17 cells based on the DNA methylation levels of STAT3, STAT5, and RORγt [112]. Inhalation of PM_2.5_ is also associated with increased apoptosis. PM_2.5_ exposure increases necroptosis inflammation through the oxidative phosphorylation pathway, which promotes the production of IL-33 and downregulation of ATP5F, NDUF, COX7A, and UQCR, inducing phosphatidylserine and the upregulation of RELA and CAPN1, and causing cell apoptosis [113]. Furthermore, PM_2.5_ induces cell autophagy by mediating the AMPK signaling pathway, oxidative stress-mediated PI3K, AKT, NOS2, and mTOR pathways, and the ATR serine/threonine kinase (ATR)-checkpoint kinase 1 axis signaling pathway (CHEK1, CHK1) which all play a major role in airway inflammation. PM_2.5_ also upregulates the expression of ATG5, LC3II, Beclin-1, IL-6, and TNF-α, further enhancing autophagy [114,115,116,117]. All these effects lead to a decline in pulmonary function, mediating the development and facilitating the exacerbation of airway-obstructive diseases through inflammation and oxidative stress. Chronic PM_2.5_ exposure would cause persistent oxidative and inflammatory damage, which would be responsible for the development and maintenance of chronic bronchitis, COPD, and asthma. [118,119]. Studies have demonstrated that increase in PM_2.5_ exposure contributes to a higher prevalence of hospitalization and severity of symptoms in children and adult patients with COPD and asthma [82]. Periodic exposure to PM has been confirmed to play a role in the prevalence of COPD and the lifetime prevalence of asthma [98,118]. The GBD study estimated that 3.1 to 3.3 million people died of COPD, whereas around 0.4 million people succumbed to asthma, worldwide in 2015. The risk estimation was based on factors such as smoking, secondhand smoke, household air pollution, ambient PM_2.5_, ozone, and occupational particulates. The underlying mechanisms of PM_2.5_-induced COPD and asthma involve induction of oxidative stress-mediated pathways and pulmonary inflammation, as illustrated in Figure 2. PM_2.5_-induced ROS upregulates the expression of the pro-inflammatory cytokines IL-6, IL-8, MCP-1, and TNF-α [120]. In densely populated regions, urban particulate matter (UPM) differs in its chemical composition, which leads to a complex mixture that indicates a site-specific variability. Some of the pollutant particles, after being emitted from the source, can be transported over long distances through the ambient air. Genes that were influenced by ultra-fine particulate matter (UFPM) in human monocytes, such as those involved in DNA repair, apoptosis, and oxidative stress, were upregulated, but interestingly, not in a dose-dependent, but rather time-dependent manner [121,122].

## 6. Cardiovascular Diseases

Epidemiological studies have shown a clear association between PM_2.5_ and cardiovascular diseases, including arrhythmia [123], cardiac arrest [124], coronary artery disease [125], heart failure [126,127], venous thromboembolism [128], and cerebrovascular disease [129]. Acute exposure has been linked to such cardiovascular diseases. Individuals presenting with myocardial infarction were more like to have been in traffic 1–2 h prior [130]. Although it is difficult to control for confounding variables such as noise and stress, correcting for activity intensity had no effect on the connection. The confounding relationship between pollution and cardiovascular health provides a prediction of cardiovascular health, noises, such as railway and air traffic, are highly connected with cardiovascular health [131]. However, with respect to considering the health effects of PM_2.5_, the confounding variables are not included to determine the PM_2.5_ effects of cardiovascular diseases. Subsequent investigations have further corroborated this link, which is independent of the mode of transportation used [132]. Epidemiological studies have associated air pollution with various end points underpinning cardiovascular conditions. Atherosclerosis in a variety of arterial beds has been linked to urban air pollution [133]. Although there is some variation, PM exposure is linked to a slight but significant increase in blood pressure (typically 5 mmHg for an interquartile increase due to PM_2.5_) [134]. The constriction or reduced vasodilation of resistance arteries, which occurs after exposure to PM, elevates blood pressure. Although not all studies have identified significant connections, exposure to PM_2.5_ and traffic (e.g., distance from major road to residential address) has been related to increased arterial stiffness [135,136,137]. Oxidative stress is the primary hierarchical response to PM exposure in humans, followed by other variables. Toll-like receptors (TLR2/TLR4) and nucleotide binding receptors are involved in this response and may be directly or indirectly activated by secondary mediators, including ROS [138,139,140]. The induction of ROS may lead to the activation of MAPK pathways, NF-κb, and AP1, which increases the synthesis of inflammatory proteins and brings about alterations in membrane permeability and mitochondrial dysfunction [141,142]. Inflammatory markers induced by PM_2.5_ directly act on the heart and induce cardiac tissue remodeling and function alteration, leading to the development of cardiac diseases. Current evidence suggests that translocated PM_2.5_ causes both systemic inflammation and sympathetic activation in the cardiovascular system. PM_2.5_ causes systemic inflammation and elevates catecholamines, leading to an acute or chronic phase response of hypercoagulable state (suppression of fibrinolysis and activation of coagulation), vasoconstriction, increase in blood pressure, endothelial dysfunction, cardiac electrical changes, imbalance of cardiac ANS [143]. Sympathetic activation increases catecholamine production, leading to endothelial dysfunction, increase in heart rate, and promotes vasoconstriction and hypertension [144]. The combined effects of systemic inflammation and sympathetic activation on their molecular targets lead to ischemic or thrombotic events, cardiac arrhythmia, and heart failure [145]. The biological pathways whereby PM_2.5_ promotes cardiovascular impairments are illustrated (Figure 3). The effects of PM_2.5_ exposure on catecholamine levels show that PM_2.5_ exposure is a major disruptor of the cardiac autonomic nervous system (ANS). Little is known about how PM_2.5_ exposure affects the cardiovascular system, thus further study is needed to discover the negative health consequences linked with PM_2.5_ exposure [145].Changes in inflammatory pathways and ER stress have been identified as the key mechanisms by which PM_2.5_ promotes IR and T2DM and activates their pathophysiological responses [143,144,145]. Modulation of visceral adipose inflammation, hepatic lipid metabolism, glucose utilization in skeletal muscle, and CC-chemokine receptor 2-dependent pathways were discovered to play a significant role in PM_2.5_-mediated IR. Furthermore, PM_2.5_ has been shown to activate unfolded protein response (UPR), an intracellular ER stress signal that governs cell metabolism and survival in vivo, by phosphorylating inositol-requiring enzyme 1 alpha in hepatic cells [144]. Additionally, UPR or UPR-mediated ER stress has been linked to inflammatory pathways and has been shown to contribute to the generation of inflammatory mediators. Inflammatory mediators might activate or spread intracellular UPR [145]. As a result of the combination between inflammation and ER stress, a positive feedback loop may form, amplifying the effects of PM_2.5_ on DM. 

## 7. Cancers

The International Agency for Research on Cancer has categorized outdoor air pollution as carcinogenic to humans based on evidence from epidemiologic and animal studies, as well as mechanistic research [146]. Studies have found a link between PM_2.5_ and the risk of lung cancer [147]. Furthermore, NO_2_ and ozone (O_3_) levels have been linked to cellular alterations associated with neoplasia: altered telomere length, expression of genes involved in DNA damage and repair, inflammation, immunological and oxidative stress response, and epigenetic effects such as DNA methylation [148]. The WHO has classified diesel engine exhaust as a carcinogen based on evidence of a relation to lung cancer. Excessive exposure to diesel exhaust or traffic pollution PM_2.5_ has also been linked to benign and malignant lung tumors in laboratory animals, colorectal cancer, and gastric cancer [147,149,150]. PM_2.5_ has been linked to both the occurrence and mortality of bladder cancer [151]. A Spanish study found a link between polycyclic aromatic hydrocarbons and diesel exhaust pollutants and bladder cancer in long-term inhabitants of an industrially polluted region [152]. Taiwanese research has linked an elevated risk of bladder cancer fatalities to ambient benzene and other hydrocarbons emitted by evaporative losses of petroleum products and motor vehicle emissions [153]. A study in Sao Paulo, Brazil, discovered a link between PM_10_ exposure and the risk of bladder cancer but not the risk of bladder cancer death [154]. The American Cancer Society’s prospective Cancer Prevention Study II, which monitored 623,048 individuals for 22 years (1982–2004), discovered that PM_2.5_ was associated with mortality from kidney and bladder cancers, and that NO_2_ levels were related to colorectal cancer mortality [155]. Benzene exposure from vehicle exhaust, particularly during pregnancy and the early years of children, has been linked to an increased risk of pediatric leukemia [156]. Prenatal exposure to PM_2.5_ may raise a child′s chance of contracting leukemia and astrocytoma [157].

Yang et al. investigated the impact of PM_2.5_ on lung cancer initiation, development, and progression in A549 and H1299 tumor cell lines [158]. They stimulated PM_2.5_-related conditions in A549 and H1299 non-small cell lung cancer cells using a PM_2.5_-exposed population (geographically based on a city with the worst air quality index compared to a control city in China). They discovered that PM_2.5_ effectively induces proliferation in H1292 tumor cells in vitro via a mechanism involving cytokines, matrix metalloprotease 1 (MMP1), and IL-1ß. They also found that MMP1 was the most upregulated gene, and a study of the epiregulin or EREG-induced signaling pathways revealed that EREG increases cell survival by modulating MMP1 expression. MMP1 has been related to cell survival and has a significant propensity to induce cancer invasion and metastasis. MMP1 and IL-1ß have been demonstrated to have roles in angiogenesis, cell invasion, and metastasis [158,159,160]. In vivo studies have shown that individuals exposed to air pollution as a result of their occupation or residential address have a higher frequency of chromosome aberrations, micronuclei in lymphocytes, and differential expression of genes involved in oxidative stress, inflammation, and DNA damage and repair [148,161,162]. However, in vivo and/or animal studies in air pollution-related lung cancer research are scarce, as are in vitro studies. Chronic exposure to traffic-related outdoor air pollution increases the incidence of lung cancer. Another study found that the chemokine CXCL13 was overexpressed in 90% of lung malignancies in highly polluted areas as compared to control areas. High CXCL13 expression was linked to advanced cancer and a bad prognosis. Furthermore, CXCL13 serum concentrations increased in mice prior to the appearance of a lung tumor detectable by microCT [163,164]. Potential molecular pathways in air pollution-related lung cancer are illustrated (Figure 4).

## 8. Neurodegenerative Diseases

Epidemiological human and animal studies support the notion that air pollution can affect the central nervous system (CNS) and lead to CNS disorders [165,166,167]. PM_2.5_ and UFPM are of special concern as they can enter the systemic circulation and spread to the brain and other organs, as well as obtain direct access to the brain via the nasal olfactory mucosa [168,169,170]. Decreased cognitive performance, olfactory problems, auditory impairments, depression symptoms, and other negative neuropsychological consequences have been reported in humans in highly polluted areas [171,172]. Controlled acute diesel exhaust (300 g/m^3^) exposure has been demonstrated to cause electroencephalogram (EEG) alterations [173]. Post-mortem examinations of highly exposed people have indicated elevations in the levels of indicators of oxidative stress and neuroinflammation [174]. Furthermore, research suggests that young people may be especially vulnerable to air pollution-induced neurotoxicity [175]. Studies in Mexico City have found heightened levels of neuroinflammatory markers in the brains of children exposed to high levels of air pollution, as well as cognitive impairments and hyperactivity in 7-year-old children linked with early life exposure to traffic-related air pollution [176]. A retrospective cohort study in Catalonia, Spain, discovered a link between air pollution (defined as residing 300 m from a highway) and the prevalence of attention deficit hyperactivity disorder (ADHD) [177]. In contrast, a major study including eight European population-based birth/child cohorts found no link between air pollution exposure and ADHD, similar to a Swedish investigation [178,179]. Experimental studies support the hypothesis that air pollution is a developmental neurotoxicant. A study by Ema, Naya, and Kato [180] indicated that developmental exposure to diesel exhaust may produce toxicity and neurotoxicity. In male mice, in utero exposure to high doses of diesel exhaust (1.0 mg/m^3^) resulted in changes in motor activity and coordination, and impulsive behavior [181]. Further research in mice revealed that postnatal injection (PND 4–7 and 10–13, for 4 h/day) of diesel exhaust particulate matter (DE-PM) (100 g/m^3^) induced alterations in GFAP expression in numerous brain areas, whereas UFPM (45 g/m^3^) caused male-specific learning and memory dysfunctions [182]. Subsequent research has revealed that embryonic DE exposure in mice impacts motor activity, spatial learning and memory, and new object recognition abilities, and alters gene expression, causing neuroinflammation and oxidative damage [183]. The effects of air pollution on the nervous system and its possible role in neurodegenerative disorders are illustrated in Figure 5.

Altogether, findings in humans and many animal models suggest that air pollution may harm the developing brain and perhaps lead to neurodevelopmental problems. Autism spectrum disorder is a prominent neurodevelopmental condition, and data from both epidemiology and controlled animal research show that PM_2.5_ may be associated with neurodevelopmental and neurodegenerative diseases [184].

## 9. Role of PM_2.5_ in Viral Infections

### 9.1. H1N1 and SARS

Epidemiological and experimental studies have revealed connections between air pollution exposure and respiratory viral infections. The H1N1 flu is a subtype of influenza A, and was first detected in the spring of 2009 in the USA; it then spread rapidly throughout the world. The H1N1 virus includes a novel mix of influenza genes that have not been detected before in either animals or humans. It was named the influenza A (H1N1) pdm09 virus because it was very distinct from the viruses that were circulating throughout the pandemic [185]. This virus, which caused a global flu pandemic in 2009–2010, was popularly known as “swine flu.” Despite the large number of studies that have been conducted to analyze the many parameters that influence susceptibility to viral infections, the processes by which inhaled oxidants might change viral pathogenesis are extremely complicated. It has been demonstrated that oxidative stress worsens the severity of viral infections. One of the most common air contaminants in cities is ozone, an elemental form of oxygen. It is a strong inducer of oxidative stress, which can lead to airway inflammation and increased respiratory morbidity [186,187]. Environmentally persistent free radicals (EPFRs) were found in PM samples taken from several cities in the United States [188]. In this regard, Lee et al. found that EPFRs associated with combustion-derived PM were crucial in increasing the severity and mortality of respiratory tract viral infections [189]. A study by Hirota et al. found that in vitro scratch injury and H1N1 influenza A exposure boosted IL-1 production in human airway epithelial cells. Several studies have been conducted with the goal of documenting the worldwide mortality effect of influenza A (H1N1) pdm09 and finding variables that explain mortality variances reported across populations [190]. Some research has concentrated on risk factors such as environmental exposure. In Brisbane, Australia, Xu et al. (2013) discovered substantial interaction impacts of PM and mean temperature on pediatric influenza [191]. When searching for possible explanations as to why some countries were harder hit by the H1N1 virus pandemic in 2009, Morales et al. (2017) highlighted the importance of monitoring environmental exposure to air pollution, which is a burden on the respiratory system and immune-compromising chronic infections [192].

In an ecologic study on air pollution conducted in China, Cui et al. (2003) discovered that patients with SARS from locations with an intermediate air pollution index (API) exhibited higher mortality compared to those with a lower API [193]. Kan et al. (2005) found similar results when they evaluated the relationship between air pollution and daily SARS mortality in the Beijing (China) population. They discovered that each 10 g/m^3^ rise in PM_10_, SO_2_, and NO_2_ levels over a 5-day moving average was associated with a relative risk of daily SARS death of 1.06, 0.74, and 1.22, respectively [194]. Cai et al. (2007) conducted ecological research in mainland China to examine the possible link between the SARS outbreak and climatic conditions and air pollution [195]. In contrast to the findings of Cui et al. (2003), they found no link between air pollution and the SARS outbreak. Although air pollution should not influence SARS-CoV survival in vitro, it may exert an effect by altering the host’s local resistance. The authors suggested that more research should be conducted on this subject [193,195].

### 9.2. Air Pollution and SARS-CoV-2 (COVID-19)

Coronavirus disease 2019 (COVID-19) is caused by a coronavirus that causes severe acute respiratory illness (SARS-CoV-2). Although a unique coronavirus illness epidemic was detected in Wuhan (China) in December 2019, the outbreak was formally confirmed as a pandemic only on 11 February 2020 [196]. A significant number of studies on SARS-CoV-2 and COVID-19 have been published in recent weeks/months. The connection between severe viral respiratory illnesses, which afflict 10–20% of the population, and air pollution is widely known [197]. Pollutants in the air, such as PM_2.5_, PM_10_, sulfur dioxide, nitrogen dioxide, carbon monoxide, and ozone, can alter airways upon inhalation, increasing susceptibility to respiratory viral infections and the severity of these illnesses [198]. In this regard, Frontera et al. (2020) recently hypothesized that an atmosphere with a high concentration of air pollutants, together with meteorological circumstances, would promote the persistence of virus particles in the air for a longer period of time, favoring indirect transmission of SARS-CoV-2 in addition to direct transmission from person to person [199]. Martelletti and Martelletti (2020) discovered that the northern regions of Italy most impacted by COVID-19 also have the greatest concentrations of PM_10_ and PM_2.5_ [200]. According to these authors, SARS-CoV-2 might find appropriate transporters in air pollution particles. Furthermore, in a linear connection, the viruses would live longer and grow more aggressive in an immune system already weakened by air pollution [200]. Individuals who live in areas with high concentrations of air pollution are more likely to acquire respiratory disorders and are more susceptible to viral infections [201,202]. Pollution wreaks havoc on the upper airway’s first line of defense, the cilia. Based on this, Conticini et al. (2020) explored whether communities living in polluted areas, such as Lombardy and Emilia Romagna, were more likely to die of COVID-19 because of their poorer previous health state induced by air pollution. The normally high concentrations of air pollution in Northern Italy have been determined to be an additional co-factor of the high level of lethality documented in that location [203]. Zhu et al. (2020) studied the connection between six air pollutant concentrations (PM_2.5_, PM_10_, CO, NO_2_, and O_3_) and daily verified COVID-19 cases in 120 Chinese cities. These contaminants were shown to have significant positive relationships with COVID-19-verified cases. However, SO_2_ levels were shown to be inversely related to the number of daily confirmed cases. Nevertheless, the findings of this study support the notion that air pollution may play a role in SARS-CoV-2 infections [204]. The findings of this study, which has been replicated in Italy (Conticini et al., 2020) and in the United States (Wu et al., 2020), suggest that PM_2.5_ leads to a large increase in COVID death rate, suggesting that persistent exposure to air pollution hinders recovery and leads to more severe and deadly types of illness [203,205]. Coccia (2020) investigated the mechanisms of COVID-19 transmission dynamics in the environment to determine a feasible approach for dealing with future epidemics comparable to coronavirus infections. Their research focused on a case study of Italy, which has one of the highest rates of mortality in the world. The findings demonstrated that increased COVID-19 transmission dynamics in certain situations were caused by two mechanisms: air pollution-to-human transmission and human-to-human transmission in a context of high population density. The two main findings were as follows: (1) the acceleration of COVID-19 transmission dynamics in North Italy was highly associated with city air pollution, and (2) cities with more than 100 days of air pollution (exceeding the limits set for PM_10_, PM_2.5_) had a very high average number of infected individuals (about 3340 infected individuals), whereas cities with less than 100 days of air pollution had a lower average number of infected individuals (about 1450 infected individuals) on April 2020 [206]. Finally, as a scientific curiosity, it is worth noting that, given the significant reduction in air pollution following the quarantine, the COVID-19 pandemic may have paradoxically reduced the total number of deaths during this period by drastically reducing the number of deaths caused by air pollution [207].

## 10. Conclusions

Molecular epidemiological research is expanding; however, studies are only being conducted in specific localities with high levels of ambient and indoor air pollution, which appear to be emphasized in the literature. This complicates the interpretation of general and local influences of air pollutants with the potential to cause significant health problems; hence, the conclusions from these studies have been deemed questionable in terms of their generalizability to other parts of the world. Furthermore, due to intrinsic and extrinsic confounders that make the generated high-throughput data difficult to interpret, such research is difficult to conduct. Laboratory experiments in air pollution research are difficult to replicate and have received little validation. Future multi-layered studies on PM_2.5_–related lung inflammation will ultimately be necessary due to the rising problem of air pollution and the increasing incidence of lung cancers, particularly in non-smokers, as well as the histopathological shift to adenocarcinoma being the predominant cancer type and cardiovascular diseases. Future research must overcome the identified challenges to enable a better understanding of the mechanisms of carcinogenicity in air pollution–related lung cancers, independent of confounding variables to determine the cardiovascular diseases. Air pollutants typically do not exist in isolation, but rather as part of a complicated network of elements that includes other environmental contaminants and exposures. These various exposures offer distinct and perhaps cumulative health concerns that have yet to be completely recognized. Finally, the inevitable climate catastrophe may provide the highest and most severe mandate to achieve both the immediate health benefits of reducing pollution exposure and the more complicated, long-term rewards of mitigating climate change in order to achieve climate change goals.

## Figures and Tables

**Figure 1 ijerph-19-07511-f001:**
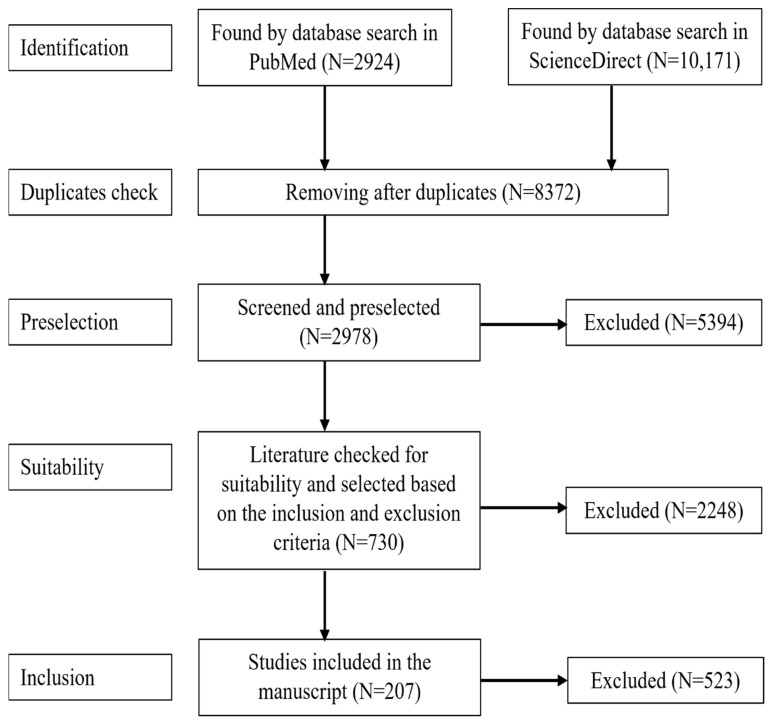
Systematic screening process of literature review.

**Figure 2 ijerph-19-07511-f002:**
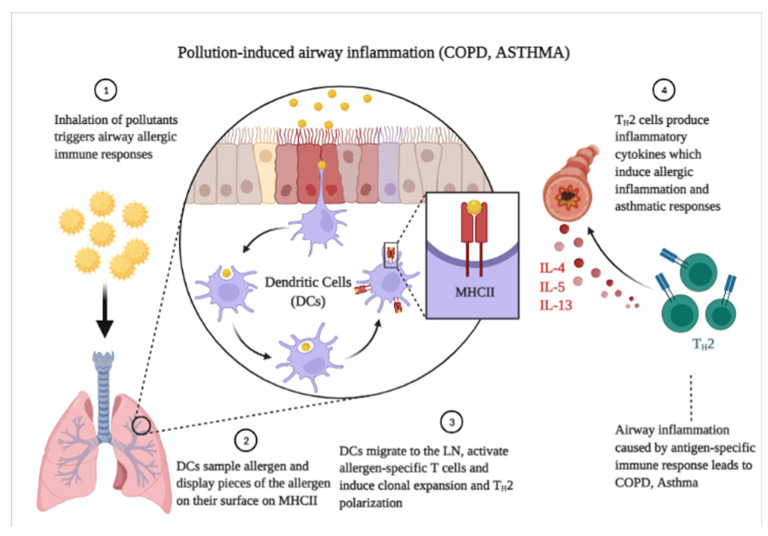
Illustration of underlying mechanisms of PM_2.5_-induced COPD and asthma.

**Figure 3 ijerph-19-07511-f003:**
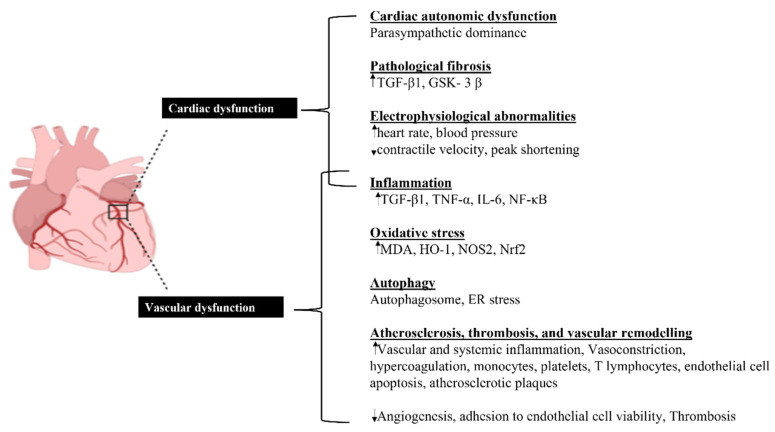
Biological pathways whereby PM particles promote cardiovascular impairments.

**Figure 4 ijerph-19-07511-f004:**
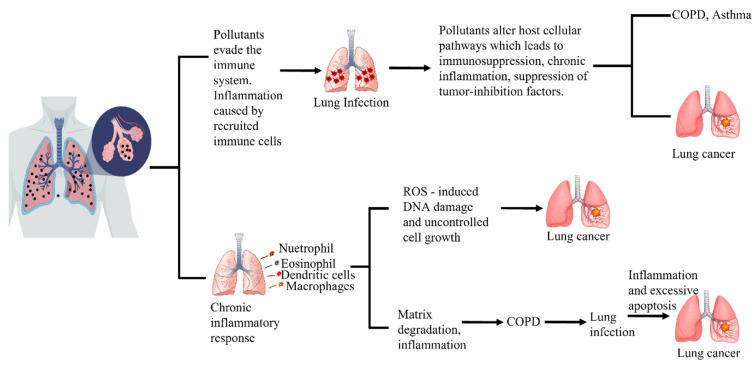
Potential molecular pathways in air pollution–related lung cancer.

**Figure 5 ijerph-19-07511-f005:**
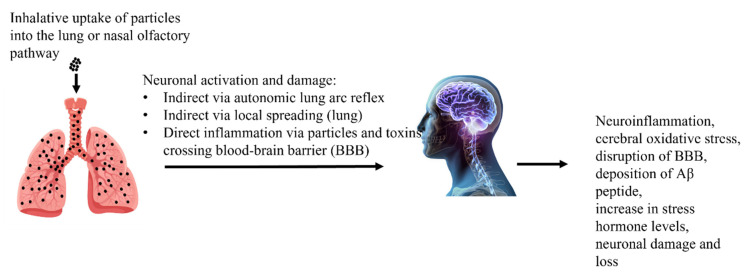
Effects of air pollution on the nervous system and its possible role in neurodegenerative disorders.

**Table 1 ijerph-19-07511-t001:** Various source of PM_2.5_ and their chemical components.

Source		Components
Natural	Biomass	Potassium (K)
	Sea spray aerosols	Sodium (Na)
	Coal burning	Aluminium (Al), Selenium (Se), Cobalt (Co), Arsenic (As)
	Soil and road dust	Aluminium (Al), Silicon (Si), Calcium (Ca)
	Volcanic dust particles and wild land fire particles	Potassium (K), Zinc (Zn), Lead (Pb)
Anthropogenic	Diesel, petrol and coal combustion	Elemental carbon (EC), Sulfates (SO_4_)
	Oil burning	Vanadium (V), Nickel (Ni), Manganese (Mn), Iron (Fe)) Organic carbon (OC)
	Heavy industry—high temperature combustion	Iron (Fe), Zinc (Zn), Copper (Cu), Lead (Pb), Nitrates (NO_3_)
	Fertilizer and animal husbandry	Ammonium (NH_4_)
	Volatile organic compound (VOC) emissions	Benzene, Ethylene glycol, Formaldehyde, Methylene chloride, Tetrachloroethylene, Toluene, Xylene, and 1,3-Butadiene

**Table 2 ijerph-19-07511-t002:** Health complications caused by PM_2.5_.

Exposure	System Affected	Health Effects	References
Short term	Cardiovascular	Increased rates of myocardial infarction and ischemia in those at risk Exacerbation of cardiac failure	[79,80,81]
	Respiratory	Increased incidence of arrhythmia Increased incidence of deep vein thrombosis Increased incidence of stroke Increased wheeze Exacerbation of asthma Exacerbation of chronic obstructive pulmonary disease Bronchiolitis and other respiratory infections Increased incidence of emergency department visits	[82,83,84,85,86]
Long term	Cardiovascular	Increased rates of myocardial infarction Accelerated development of atherosclerosis Increased blood coagulability	[87,88,89]
	Respiratory	Increase in systemic inflammatory markers Increased incidence of pneumonia Increased incidence of lung cancer Impaired lung development in children Development of new asthma	[90,91,92,93,94,95]
	Reproductive	Increased incidence of preterm birth Increased incidence of low birth weight	[89,93]
	Brain	Increased risk of Alzheimer’s Increased risk of Parkinson’s Increased risk of neurodegenerative diseases	[96,97]

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
