# Peer review of "Recent Insights into Particulate Matter (PM2.5)-Mediated Toxicity in Humans: An Overview"

_ijerph, 2022, doi:10.3390/ijerph19127511_

Round 1

Reviewer 1 Report

The scope of the paper should be defined clearly in the abstract – is it focused on all PM particles or specifically in PM2.5 particles? Indeed, while in the tile “Recent insights into particulate matter (PM2.5)-mediated toxicity in humans: an overview” there is a specific mention to PM2.5 particles, in the abstract particles ranging from 2.5 to 10 are mentioned and it is written that “This review article summarizes the basic evidence on PM-mediated toxicity”. In fact, throughout the paper there is emphasis to PM2.5, but there are also mentions to PM10 and to UFPM in some parts of the paper.

Line 29 – Where is written “air pollution combined with particulate matter (PM)” the authors should consider that PM itself is an air pollutant and hence a form of air pollution.

A clear definition of PM should be provided, since the issues presented in the introduction are confusing: the effects induced by gaseous pollutants (or by pollutants in general?) are mentioned and suddenly a presentation of PM appears. What was the purpose of mentioning gaseous pollutants? 

Line 50 – Where is written “as they can navigate through bronchioles and escape lung defence mechanisms”, which ones? All PMs? The authors could mention that smaller particles can cross natural barriers easily.

Line 70 – It is mentioned “These processes lead to oxidative stress”, however when there is inflammation in lungs the inflammation itself leads to oxidative stress. This could be added in the text.

Line 71 – Both sentences “The relationships between air quality and health are disproportional; the particles of air pollutants adversely affect the productivity of agricultural workers, land miners, and sewage workers” are unclear.

Lines 83 and 92 are repeating. In all this section the text focusing sources and composition is disorganized.

Line 103 – What is the meaning of “secondary generation of air pollution”? Briefly explain it in the text.

Line 120—Spell out “ANS”; the same for “CNS” in line 291 and “EEG” in line 298.

Line 177 – The text gives the idea that the infectious diseases lead to a decline in pulmonary function, mediating the development and facilitating the exacerbation of airway-obstructive diseases. To my knowledge diseases such as COPD can come from chronic toxic insults and these ones lead to inflammation and/or oxidative stress which in fact illustrated by the authors in figure 1.

Line 242 – Please check the sentence “Many studies have found a link between PM2.5 and PM with an aerodynamic diameter of 10 µm (PM10) …”. It is unclear.

Line 329 – In this context the term “genealogy” is unclear.

Line 360 – Where is written “…discovered that patients with SARS from locations with intermediate air pollution index (API) exhibited an 84% greater probability of dying compared to patients from regions with low APIs”, I wonder if the previous presence of PM2.5-induced  conditions (such as lung inflammation/oxidative stress) simply did not exacerbate SARS’ effects- especially considering that “It has been demonstrated that oxidative stress worsens the severity of viral infections (line 342); this could be mentioned in the text since only associations between PM levels and mortality are being presented. On the other hand, in Line 374 it is mentioned “it may exert an effect by altering the host’s local resistance” - this is the only potential association between PM2.5 and viral infection. I believe that this could be developed in this section.  

Reviewer 2 Report

This paper provides an extensive list of studies reporting associations between PM2.5 and various health endpoints. However, the paper does not provide sufficient space to discussing the significance, statistical or otherwise of the study results presented and the main methodological issues, if any, associated with the studies. It would also be helpful to have some indication of what studies the authors think should be performed to reduce uncertainties in determining the importance of PM2.5 components for the health components mentioned.

The discussion of the sources and composition of ambient particles is lacking in many places. Perhaps a table outlining natural and anthropogenic sources separately with examples of major components would be better than the discussion in the text.

Minor 

l. 48 – It would be worthwhile to note that the size designations for PM refer toa aerodynamic diameter, not physical diameter.  

l. 56 – Explain what is meant by saying PM2.5 is characterized by large surface area. Compared to what, PM10-2.5?

l. 56-57 – Do the authors mean PM2.5 accumulates during atmospheric stagnation?

l. 59 - delete “or any gas phase species”.

l. 60 – might be easiest to delete such as elemental carbon as EC is not known to originate from dust storms. 

l. 71-73 – “Airborne particles adversely…”.   It’s not clear what the authors are trying to say. Ambient PM affects the health of everyone.

l. 81 – You already mentioned that sources of PM are both natural and anthropogenic. Delete.

References are missing for long term pulmonary disease in Table 1.

l. 178-11 – There should be references given here.

l. 204-205 – Some more detail about why confounding is ruled out should be given here. There are many open questions about the role of confounding in epidemiologic modeling.      

Reference 190 needs citing information.

Reviewer 3 Report

The authors should be commended on the depth and breadth of their literature search. However, the clarity and organization of the manuscript should be improved. The language is often confusing and there are many instances of repetitive sentences. Additionally, for the sections on the health effects of PM, it seems like a repetition of each study's findings rather than a synthesis of the findings overall. Organizing the studies and aggregating the findings to tell the overall narrative would greatly improve clarity, readability, and impact. 

Reviewer 4 Report

For several years air pollution has been a major environmental health risk, moreover WHO reported that approximately 730 million people die each year because of exposure on polluted air. Air pollution consist of gases, particles, and biological components, so it is a mixture of substances called particulate matter (PM). The toxic effect of PM depends on the size, superficial area, and composition. PM can be classified by its size as P<10 (particles ≤10 μm in diameter), PM2.5 (particles ≤2.5 μm in diameter), also called fine particles, and PM0.1 (particles ≤0.1 μm in diameter), called ultrafine particles. The size of PM is directly related to the potency of them to cause health problems.

The review is focused on insight into particulate matter (PM2.5)- related toxicity in humans. PM with diameter of 2.5um and smaller can penetrate the lower tract and have a harmful direct and indirect effects on various organs and tissues in humans.

The structure of the review paper is correct. Authors very thoroughly described the effect of PM on cardiovascular, respiratory, and neurological diseases. All information is based on 192 very recent and well citied literature positions, mostly groups investigating the effect of air pollution on human health. Four figures illustrate mechanisms of PM2.5-induced COPD and asthma, biological pathways of PM2.5-promoted cardiovascular impairments, potential molecular pathways in air pollution–related lung cancer and the effects of air pollution on the nervous system and its possible role in neurodegenerative disorders. Finally, authors discussed the role of PM2.5 in viral infections including influenza A (H1N1), SARS-CoV-2 (COVID-19)

The review paper is very well written and can help to challenge air pollution-related increasing incidences of lung cancers, particularly in non-smokers, as well as other cardiovascular and neurological disorders.

Round 2

Reviewer 2 Report

Natural sources do not appear to be represented in table 1. Be sure to clearly identify which sources are natural and which are anthropogenic.

The authors should at least mention up front what criteria they used to judge the scientific soundness of the articles they've included.

Reviewer 3 Report

The manuscript is a marked improvement from the initial draft in clarity and impact. There are still some English language errors, however, that does not detract from the manuscript as a whole. The following should be addressed though before publication:

- Table 1: While this table is quite informative, the difference between natural and anthropogenic sources is not clear. Additionally, the table could be better organized to make the sources and pollutants clearer.

- Page 4, sentence starting, "Recently, increasing evidence has shown..." is a run on sentence with multiple dependent clauses and should be rewritten.

- Page 9: much of this paragraph is on the connection between PM2.5 and diabetes, not cardiovascular health. Thus, it should not be in this section

- Page 15: "Air pollutants typically exist in isolation..." should be rewritten as "Air pollutants typically do not exist in isolation..."
